# CoEdIT: Text Editing by Task-Specific Instruction Tuning

**Vipul Raheja**♣     **Dhruv Kumar**♣     **Ryan Koo**◇     **Dongyeop Kang**◇

♣Grammarly     ◇University of Minnesota

♣`first.last@grammarly.com`     ◇`{koo00017, dongyeop}@umn.edu`

## Abstract

We introduce CoEdIT, a state-of-the-art text editing system for writing assistance. CoEdIT takes instructions from the user specifying the attributes of the desired text, such as "*Make the sentence simpler*" or "*Write it in a more neutral style,*" and outputs the edited text. We present a large language model fine-tuned on a diverse collection of task-specific instructions for text editing (a total of 82K instructions). Our model (1) achieves state-of-the-art performance on various text editing benchmarks, (2) is competitive with publicly available largest-sized LLMs trained on instructions while being ~60x smaller, (3) is capable of generalizing to unseen edit instructions, and (4) exhibits abilities to generalize to composite instructions containing different combinations of edit actions. Through extensive qualitative and quantitative analysis, we show that writers prefer the edits suggested by CoEdIT, relative to other state-of-the-art text editing models[1].

## 1  Introduction

Large language models (LLMs) have made remarkable progress toward generating coherent text in a wide variety of tasks and domains to support writing assistance (Du et al., 2022a; Mallinson et al., 2022; Schick et al., 2023), such as grammatical error correction (Wu et al., 2023), text simplification (Štajner et al., 2022), paraphrasing (Chowdhury et al., 2022), and style transfer (Reif et al., 2022). One of the emergent abilities of LLMs is the capability to generalize to unseen tasks by following new or composed instructions. Instruction-tuning, where LLMs are fine-tuned on a collection of tasks phrased as instructions, makes the models more adept at interpreting and following instructions, reducing the need for few-shot exemplars (Sanh et al., 2022; Ouyang et al., 2022b; Wei et al., 2022; Chung et al., 2022b).

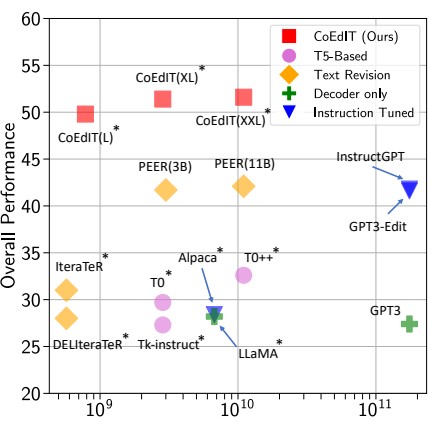

Figure 1: Model comparison according to training parameters vs. average performance across all text editing benchmarks reported in Tables 2 and 11. Publicly available models are denoted with (*).

Text editing is a complex task because human writers cannot simultaneously grasp multiple demands and constraints of the task and tend to iterate and revise their work multiple times (Flower, 1980; Collins and Gentner, 1980; Vaughan and McDonald, 1986). This poses a significant challenge for intelligent writing assistants.

In this work, we aim to improve the capabilities of instruction-tuned models for text editing by leveraging instruction-tuning from diverse tasks of text editing benchmarks. While multiple previous works have attempted to develop general-purpose text editing models using LLMs, they are either not trained with instruction-tuning (Du et al., 2022c; Kim et al., 2022), trained on much smaller models or not trained on task-specific datasets (Mallinson et al., 2022; Schick et al., 2023), or are not publicly available (Schick et al., 2023), which limits their effectiveness, performance, or usability.

We introduce CoEdIT, a text editing system designed to provide writing assistance with a natural language interface. A user can employ CoEdIT by providing natural language instructions such as "*Paraphrase the sentence*" or "*Fix the grammar*". Our experiments demonstrate that fine-tuning in-

---

[1]Code, data, and models available at `https://github.com/vipulraheja/coedit`

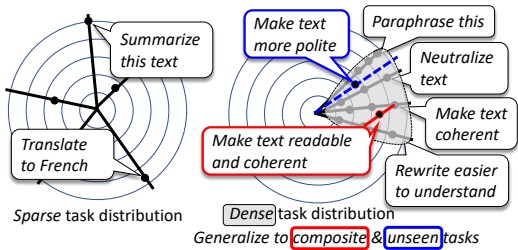

Figure 2: General-purpose (left) vs Task-specific (right) Instruction Tuning.

structions for specific tasks is more effective than multi-task learning and general-purpose instruction tuning. We conjecture that task-specific instructions increase the density of the instruction space, reinforcing the complementary effects of multiple tasks and facilitating their generalization to composite and new text editing tasks, as shown in Fig. 2.

To build CoEdIT, we fine-tune a pre-trained sequence-to-sequence model on a parallel corpus of instruction-based 82K input-output pairs. The inputs and outputs are sourced from publicly available corpora for different text editing tasks, and the instructions are constructed based on rules that introduce lexical and semantic variations.

Our main contributions are as follows:

- We achieve state-of-the-art performance on multiple text editing tasks: grammatical error correction, text simplification, sentence fusion, iterative text editing, and three stylistic editing tasks (formality style transfer, neutralization, and paraphrasing).

- We find that even our smallest instruction-tuned model outperforms other supervised text editing models, instruction-tuned models, and general-purpose LLMs with nearly 60x greater parameters, on both manual and automatic evaluations.

- CoEdIT generalizes well to new, adjacent tasks not seen while fine-tuning, as well as composite instructions with multiple task specifications.

- Our data and models will be publicly available.

## 2 Related Work

**Large Language Models for Text Editing** In general, our work is related to many prior works that leverage LLMs; for instance, finetuning T5 (Raffel et al., 2020a) on pairs of original and edited text (Faltings et al., 2021; Reid and Neubig, 2022; Mallinson et al., 2022; Du et al., 2022a,b; Kim et al., 2022). However, these aforementioned works are either not based on instruction tuning, use different modeling techniques such as tag-based se-

quence labeling, or are not general enough to work on multiple text editing tasks. Moreover, several LLMs are trained to solve specific tasks only, such as grammar errors (Mallinson et al., 2022; Fang et al., 2023), text simplification (Štajner et al., 2022), paraphrase generation (Chowdhury et al., 2022), or style transfer (Reif et al., 2022), which limits their generalizability.

**Instruction Tuning for Writing Assistance** Explicitly teaching models how to follow natural language instructions is closely related to recent work for fine-tuning models using large datasets of human-written instructions (Wei et al., 2022; Mishra et al., 2022; Sanh et al., 2022; Ouyang et al., 2022a; Wang et al., 2022; Iyer et al., 2022; Bach et al., 2022; Longpre et al., 2023). Recently, advanced data augmentation and instruction tuning, starting with the Flan models (Chung et al., 2022b), have shown that strong results stem both from the larger and more diverse set of tasks. Additionally, enriching task diversity and balancing task sources (Sanh et al., 2022) are shown to be critical to performance, suggesting instruction-tuned models offer a more computationally-efficient starting checkpoint for downstream applications, corroborating Liu et al. (2022) and Aribandi et al. (2022).

On instruction tuning for writing assistance, our work is closely related to PEER (Schick et al., 2023), who fine-tuned T5-based LLMs by following user-provided text-editing plans to perform the said edits. There are a few significant differences in our approach compared to PEER. While PEER attempts to either create or leverage a user-provided *plan*, realize the *edits* conditioned on the plan, and try to *explain* the plan, we focus only on the *plan* and *edit* parts of the pipeline. Even when it comes to handling editing plans in the form of natural language instructions, our work focuses on edits that do not add new information. Therefore, we compare our models only against PEER-Edit models.

Finally, no prior works, to the best of our knowledge, have investigated the ability of instruction-tuned LLMs for text editing to generalize to composite instructions.

## 3 CoEdIT

### 3.1 Training Dataset

Our dataset creation is based on the ITERATER+ dataset proposed by Kim et al. (2022) who combined datasets from various text editing tasks (See Table 1). Their work, in turn, is based on Du et al.

| Edit Intention | Datasets | Size | Example Input | Example Output |
|---|---|---|---|---|
| FLUENCY | NUCLE-14, Lang-8, BEA-19 | 20k | *Fix the grammar:* When I grow up, I start to understand what he said is quite right. | When I grew up, I started to understand what he said was quite right. |
| COHERENCE | DiscoFuse | 11k | *Make this text coherent:* Their flight is weak. They run quickly through the tree canopy. | Their flight is weak, but they run quickly through the tree canopy. |
| CLARITY (Simplification) | NEWSELA, WikiAuto, WikiLarge, ParabankV2, ITERATER-CLARITY | 13k | *Rewrite to make this easier to understand:* A storm surge is what forecasters consider a hurricane's most treacherous aspect. | A storm surge is considered a hurricane's most dangerous aspect. |
| STYLE (Paraphrase) | ParabankV2 | 15k | *Paraphrase this:* Do you know where I was born? | Do you know my birthplace? |
| STYLE (Formalize) | GYAFC | 12k | *Write this more formally:* omg i love that song im listening to it right now | I love that song and I am listening to it at this moment. |
| STYLE (Neutralize) | WNC | 11k | *Write in a more neutral way:* The authors' exposé on nutrition studies. | The authors' statements on nutrition studies. |

Table 1: Example data instances in the COEDIT dataset. Instructions in the inputs are *italicized*.

(2022b), who categorized each edit into MEANING-CHANGED or NON-MEANING-CHANGED. Edits that belong to the latter group are further assigned to FLUENCY, COHERENCE, CLARITY, or STYLE. The aforementioned taxonomy of edit intents from ITERATER reflects writers' general intention behind their revision, providing more in-depth information than just superficial edit operations, such as ADD and DELETE.

Similar to Kim et al. (2022), our work focuses on non-MEANING-CHANGED edits. We consider those edits to be ones that do not add new information or perform fact updates. Since the STYLE edits are quite subjective in nature, we allow for the possibility of meaning change so as to fulfill the needs of making stylistic edits, but we constrain the editing tasks to ensure the edited texts are semantically similar to the sources, but not to the extent of adding new information or fact updates. With this in mind, we expand the STYLE edit intention category from ITERATER+ to include three new sub-intentions: *Paraphrasing*, *Formality Style Transfer* (or *Formalization*), and *Neutralization*.

The aforementioned ITERATER dataset taxonomy lends itself conveniently to be articulated as natural language instructions and allows us to naturally formulate them into instructional prompts (See Table 1). We rewrite each edit intention as a set of natural language instruction prompts to create the COEDIT dataset. To allow models to adapt to linguistic variations of the instructions, we also include paraphrases of the instruction templates, e.g., instead of "*Write*" we also use "*Generate*" or "*Rewrite*," or instead of "*Paraphrase the text*" we use "*Rewrite the text with different wording*," and so on. For each task, we develop a variety of such diverse instructional prompts and ran-

domly sample an instruction from the aforementioned group of task-specific instruction candidates to be pre-pended to the source in order to form an `<instruction: source, target>` data pair. We provide the full list of our instructional prompts in §C. In total, our training dataset consists of around 82K `<instruction: source, target>` pairs. We keep the original train-validation-test splits consistent as the original datasets but diversify the train and validation splits with the paraphrasing augmentations. The details of datasets and instructions used to train our models are described in §A.

### 3.2 Text Editing Models

We fine-tune different versions of pre-trained FLANT5 (Chung et al., 2022a) models on the COEDIT dataset. Specifically, we use FLANT5-L (770M parameters), FLANT5-XL (3B parameters), FLANT5-XXL (11B parameters) models, which are henceforth referred to as COEDIT-L, COEDIT-XL, and COEDIT-XXL respectively. The training details are summarized in §D.

## 4 Experimental Setup

We conduct experiments to determine if a standard instruction-tuned language model fine-tuned using task-specific data can improve text editing performance and if it can further generalize into a general-purpose text editing model capable of following human-written instructions and handling a wider array of editing tasks, such as unseen and composite instructions. Specifically, we aim to answer the following research questions:

- **RQ1**: Can COEDIT follow text editing instructions and perform high-quality edits across a wide variety of tasks?

- **RQ2**: Is CoEdIT generalizable to perform high-quality edits for new types of text editing instructions?
- **RQ3**: Does CoEdIT make the writing process more efficient and effective for human writers?

We answer these questions via quantitative analyses of model outputs (Section 5) and via qualitative analyses and human evaluations of model outputs (Section 6). Further, we investigate RQ2 along two dimensions: (1) generalization to composite instructions containing combinations of multiple different kinds of edits and (2) out-of-domain generalization to instructions with new task requirements on previously unseen data.

### 4.1 Models

**No-Edits Baseline** We first evaluate a no-edits baseline, where the output is simply a copy of the source input without the instruction. This strategy performs reasonably well on tasks where the target output largely overlaps with the input (e.g., GEC).

**Supervised Text Editing Models** We also evaluate existing LLMs for text editing that are not fine-tuned with instruction-specific data. Specifically, to understand the effect of task-specific fine-tuning, we evaluate against T5[2] (Raffel et al., 2020b) models as primary alternatives of our FLAN-T5 models. We also compare our models against ITERATER (Du et al., 2022b) and DELITERATER (Kim et al., 2022), which have shown strong performance on a variety of text editing tasks.[3]

**Instruction-tuned LLMs** A major group of our comparisons is against instruction-tuned LLMs:
- Our main comparison is against **PEER** (Schick et al., 2023), which is primarily based on the *LM Adapted* variant of T5. As the focus of our work is on improving revision quality (Section 2), we compare against PEER-EDIT (both 3B and 11B versions).
- **T0**, **T0++** (Sanh et al., 2022) and **T**$k$**-Instruct** (Wang et al., 2022), which are all initialized from the *LM Adapted* variant of T5, and fine-tuned using PromptSource (Bach et al., 2022), and Super-NaturalInstructions (Wang et al., 2022) datasets, respectively.

- **Alpaca** (Taori et al., 2023) is an instruction-tuned version of the LLaMA-7B model (Touvron et al., 2023) trained on 52K instruction-following demonstrations generated by GPT3.
- We also compare **InstructGPT** (Ouyang et al., 2022a), a variant of GPT3 fine-tuned via reinforcement learning on a large dataset of instructions and human-written outputs.[4]
- **GPT3.5** (henceforth referred to as **ChatGPT**), is an improved version of InstructGPT optimized for chat. We utilize OpenAI's API for all inference tasks.[5]
- GPT3 also offers a text **Editing API**[6] (we refer to as **GPT3-Edit**), which is usable for editing tasks rather than completion, making it directly comparable to the tasks we train CoEdIT on.

**Large-Pretrained Decoder-only Models** We compare against LLMs with no instruction tuning in two settings – zero-shot and few-shot (details in Section 5.1):
- The 175B **GPT3** (Brown et al., 2020) model that is not instruction-tuned demonstrates strong general-purpose text revision capabilities.
- **LLaMA** (Touvron et al., 2023) is Meta AI's general-purpose language model trained only on publicly available data. We utilize the 7B model due to computing constraints.

Outputs of all models were generated using greedy decoding unless specified otherwise.

### 4.2 Test Datasets

To assess the editing capabilities of CoEdIT, we perform evaluations on standard test sets sourced from a variety of text editing task benchmarks, most notably, EDITEVAL (Dwivedi-Yu et al., 2022). Owing to the overlap of our work with PEER, we keep our evaluation datasets and evaluation metrics as close to theirs as possible for consistency: We used JFLEG (Napoles et al., 2017) for grammatical error collection, TurkCorpus (Xu et al., 2016) and ASSET (Alva-Manchego et al., 2020) for text simplification, Coherence split of ITERATER (Du et al., 2022b) and the DISCOFUSE dataset (Geva et al., 2019) for coherence, and ITERATER (Du et al., 2022b) for iterative text revision. For Style-related edits, we used GYAFC (Rao and Tetreault, 2018) for formality style, WNC (Pryzant et al., 2020) for neutralization, and MRPC (Dolan and

---

[2]The original T5 model cannot continue text well due to its infilling pre-training objective. Hence, similar to Schick et al. (2023), we evaluate its *LM Adapted* versions (Lester et al., 2021), which are trained with a language modeling objective.

[3]We are unable to make full comparisons against EdiT5 (Mallinson et al., 2022) and PEER (Schick et al., 2023) as the models are not publicly available.

[4]We use `text-davinci-003`

[5]We use `gpt-3.5-turbo`

[6]We use `text-davinci-edit-001`

Brockett, 2005), STS (Cer et al., 2017), and QQP for paraphrasing. Detailed descriptions of each dataset and its evaluation metrics are in §B.

# 5 Quantitative Results

## 5.1 Text Editing Performance

Table 2 helps us answer **RQ1** by comparing the performance of COEDIT to other models across various text editing tasks. We first present results from the more well-known evaluation sets here and present additional results (i.e., sub-tasks and additional datasets) in Table 11.

We segregate the models into seven groups. The first group (a) consists of the copy baseline and T5-LARGE baseline fine-tuned with prefix-tuning (each data point is prefixed with task-specific tags rather than instructions), while the second group (b) consists of instruction-fine-tuned T5-based models on non-text-editing tasks. We find that COEDIT substantially outperforms these models across all tasks.

The next two groups (c, d) show different LLMs varying from 7B to 176B parameters in size, evaluated in a zero-shot setting. Those in group (c) are decoder-only models, while those in group (d) are instruction-tuned. We find that COEDIT outperforms all LLMs comparable to its model size (e.g., Alpaca and LLaMA) across all tasks, as well as on most tasks compared to models several times larger, such as ChatGPT and InstructGPT. This indicates that current general-purpose and instruction-tuned models are underfitted, and it is beneficial to densify the task/instruction space rather than to scale model size.

Although models such as Alpaca and T5-based models (T*k*-instruct, T0, T0++) have previously shown strong capabilities for zero-shot tasks, they show weaker performance compared to COEDIT. We also see that the decoder-only models (e.g., GPT3 and LLaMA) often repeat the input for more complex tasks, such as ones under the *Style* intent group. This can be attributed to difficulty understanding the prompted task, resulting in the models either repeating the input sentence or generating a continuation unrelated to the task.

---

[7] Since PEER had several scores missing, and due to the high scores of paraphrasing transfer, for fairness, it was left out of the Overall score calculations. For results with multiple metrics, the best-performing method is calculated by taking the average. For the MRPC average, we subtract the Self-BLEU score from 100 since lower is better.

Next, in the fifth group (e), we evaluate the LLMs under a few-shot setting. As mentioned in Section 4.1, we conduct these experiments in a 4-shot evaluation setting, where example inputs were constructed by randomly sampling four inputs for each task from the COEDIT dataset such that all examples chosen would fit in the input window for all models as seen in (Brown et al., 2020). The input sentence and its corresponding revised reference were pre-pended to the instructional prompt. We conduct few-shot evaluations for decoder-only LLMs (GPT3) and three instruction-tuned LLMs (InstructGPT, ChatGPT, and Alpaca). Outputs of all models were generated using greedy decoding unless specified otherwise.

We observe that giving specific examples improves performance in all models for all tasks except MRPC for GPT3. This may be because GPT3 still exhibits some similar behavior in repeating its generations continuously, resulting in a low BLEU score but low semantic similarity as well. We don't present any experiments for GPT3-Edit under the few-shot setting, as scores tended to stay the same across all tasks – implying that GPT3-Edit may not have as good in-context learning capabilities. Overall, we find that even our smallest 770M parameter model is competitive against LLMs evaluated in a few-shot setting in most tasks.

In the final group (f), we compare our models against task-specific text editing models such as ITERATER, DELITERATER, and PEER. ITERATER and DELITERATER perform comparatively worse than the scores reported in the original paper as we present different and more difficult inputs, only pre-pending instructions to the inputs while ITERATER and DELITERATER were trained with task-specific tags. Furthermore, they were trained using BART and Pegasus, respectively, both of which have a summarization pre-training objective, and were not trained to follow instructions. On average, COEDIT beats PEER across all reported evaluations except the ITERATER benchmark. This can primarily be attributed to the difference in task-specific fine-tuning since PEER uses Wikipedia as the source of instructional edit data.

## 5.2 Ablation Studies

Table 3 shows the performance of various baselines, which we discuss in detail in this section.

**Instruction Tuning.** To understand the effectiveness of instruction-tuning, we fine-tune the 3B pa-

| | Model | Size | Overall | IteraTeR | Fluency | Clarity | Coherence | Style | | |
|---|---|---|---|---|---|---|---|---|---|---|
| | | | | ITERATER↑ | JFLEG↑ | ASSET↑ | DiscoFuse-Wiki↑ | GYAFC(↑/↑) | WNC(↑/↑) | MRPC(↓/↑) |
| (a) | COPY | - | 27.6 | 29.8 | 26.7 / 40.5 | 20.7 | 30.8 | 17.6 / 10.6 | 31.85 / 0 | 47.4 / 100 |
| | T5-LARGE | 770M | 24.7 | 21.1 | 32.7 / 22.9 | 35.8 | 28.01 | 30.9 / 4.89 | 13.2 / 0 | 27.6 / 62.8 |
| (b) | T0* | 3B | 29.7 | 26.1 | 42.2 / 36.1 | 33.2 | 32.4 | 37.9 / 39.3 | 19.4 / 0 | 28.3 / 84.1 |
| | Tk-INSTRUCT* | 3B | 27.3 | 21.0 | 35.2 / 26.8 | 36.9 | 28.9 | 35.7 / 43.01 | 24.2 / 0.1 | 20.4 / 48.9 |
| | T0++* | 11B | 32.6 | 31.5 | 39.4 / 40.5 | 33.1 | 35.5 | 36.8 / 43.7 | 21.2 / 0 | 42.9 / 94.9 |
| (c) | LLAMA | 7B | 28.2 | 30.1 | 27.7 / 3.34 | 21.8 | 31.1 | 18.8 / 89.1 | 31.9 / 0 | 5.29 / 64.2 |
| | GPT3 | 175B | 27.4 | 23.3 | 38.1 / 2.8 | 34.8 | 26.2 | 36.6 / 87.9 | 23.4 / 0 | 0 / 51.7 |
| (d) | ALPACA | 7B | 28.4 | 30.4 | 28.5 / 6.4 | 22.0 | 31.1 | 18.9 / 94.4 | 31.9 / 0 | 0 / 77.9 |
| | GPT3-EDIT | 175B | 41.8 | 36.1 | 52.4 / 50.6 | 32.9 | 54.0 | 35.7 / 52.3 | 50.7 / 17.1 | 22.6 / 98.7 |
| | INSTRUCTGPT | 175B | 41.6 | 32.6 | 62.4 / 57.2 | 44.6 | 47.4 | 47.8 / 98.2 | 33.7 / 0.1 | 16.03 / 98.9 |
| | CHATGPT | - | 36.9 | 28.2 | 57.6 / 49.4 | 45.9 | 40.2 | 40.7 / 99.6 | 28.5 / 0.1 | **13.4 / 99.0** |
| (e) | ALPACA (FS) | 7B | 30.0 | 30.8 | 33.03 / 11.3 | 23.2 | 33.1 | 20.6 / 95.4 | 32.04 / 0 | 0.1 / 66.7 |
| | GPT3 (FS) | 175B | 38.4 | 32.4 | 50.1 / 4.1 | 39.2 | 45.1 | 43.1 / 97.2 | 36.7 / 0 | 0 / 14.5 |
| | INSTRUCTGPT (FS) | 175B | 45.1 | 36.2 | 64.5 / 55.7 | **46.3** | 55.2 | 47.3 / 98.8 | 42.8 / 0 | 15.9 / 99.5 |
| | CHATGPT (FS) | - | 40.1 | 30.8 | 58 / 50.6 | 45.4 | 51.2 | 42.3 / 99.6 | 34.1 / 0 | 13.3 / 96.1 |
| (f) | ITERATER | 570M | 31.0 | 32.8 | 35.9 / 34.3 | 21.8 | 30.1 | 22.7 / 54.1 | 34.2 / 0 | 40.5 / 97.8 |
| | DELITERATER | 570M | 28.0 | 29.9 | 27.5 / 31.2 | 21.2 | 32.2 | 18.1 / 57.8 | 31.9 / 0 | 39.1 / 100 |
| | PEER-3B* | 3B | 41.7 | 37.1 | 55.5 / 54.3 | 30.5 | - | - | 53.3 / 21.6 | - |
| | PEER-11B* | 11B | 42.1 | **37.8** | 55.8 / 54.3 | 29.5 | - | - | 54.5 / 22.8 | - |
| (g) | **COEDIT-L** | 770M | 49.8 | 35.2 | 62.4 / 59.3 | 42.4 | 75.3 | 54.6 / 98.0 | 69.3 / 46.4 | 23.3 / 99.1 |
| | **COEDIT-XL** | 3B | 51.4 | 36.6 | 64.5 / 60.7 | 42.2 | **80.5** | **55.1 / 98.3** | 70.4 / 48.8 | 21.3 / 99.6 |
| | **COEDIT-XXL** | 11B | **51.5** | 37.1 | **65.0 / 61.5** | 41.7 | 78.6 | 55.1 / 97.2 | **71.0 / 51.4** | 21.8 / 99.0 |

Table 2: Comparison of COEDIT against various baselines: **(a)** copy baseline and T5-LARGE baseline with task-specific prefixes (i.e. <gec>, <clarity>, etc.) **(b)** T5-based models, **(c)** Decoder-only LLMs (zero-shot), **(d)** Instruction-tuned LLMs (zero-shot), **(e)** Few-shot evaluations of pre-trained LLMs, **(f)** SOTA text editing models, and, **(g)** Variants of COEDIT models (our work). The first score for each task (excluding MRPC style task) is SARI. The second scores for Fluency, GYAFC, and WNC are GLEU, Formality Transfer accuracy (%), and EM. For MRPC, the first score is Self-BLEU, while the second score is semantic similarity. The best-performing models[7] for each dataset are highlighted in boxes. Results with (*) are ones reported in prior works. (FS) denotes few-shot evaluation. Results on other datasets are in Table 11.

| | Model | Size | IteraTeR | Fluency | Clarity | Coherence | Style | | |
|---|---|---|---|---|---|---|---|---|---|
| | | | ITERATER↑ | JFLEG↑ | ASSET↑ | DiscoFuse-Wiki↑ | GYAFC(↑/↑) | WNC(↑/↑) | MRPC(↓/↑) |
| | COEDIT-XL | 3B | 36.6 | 64.5 / 60.7 | 42.2 | 80.5 | 55.1 / 98.3 | 70.4 / 48.8 | 21.3 / 99.6 |
| (a) | T5-XL (prefix) | 3B | 34.3 | 61.8 / 58.6 | 41.0 | 71.4 | 50.7 / 94.6 | 62.7 / 33 | 30.6 / 87.4 |
| (b) | FLANT5-XL | 3B | 30.2 | 28.3 / 41.3 | 25.5 | 37.9 | 25.0 / 27.8 | 33.5 / 0.0 | 46.6 / 92.5 |
| (c) | COEDIT-XL-R | 3B | 33.9 | 63.8 / 60.2 | 36.2 | 69.7 | 52.6 / 48.7 | 69.2 / 25.4 | 35.4 / 92.6 |

Table 3: Ablation results for COEDIT to evaluate the impact of **(a)** instruction tuning **(b)** task-specific training, and **(c)** quality of instructions. The scores from left to right follow exactly as Table 2.

rameter T5 model (T5-XL) and compare it with COEDIT-XL, its FLANT5 counterpart on the same training and validation sets. The only change is that the instructional prompts for the training datasets are replaced by task-specific prefixes. Specifically, the $82k$ <instruction: source, target> pairs in the training dataset used to train the COEDIT models were modified to <task: source, target>[8]. We observe (Table 3(a)) that the instruction-tuned COEDIT models consistently outperform prefix-tuned T5 models, showing the ef-

fectiveness of instruction-tuning over prefix-tuning.

**Task-Specific Training.** A core contribution of this work is to push the performance of small- (<1B parameters) to medium-sized (1-10B parameters) LLMs for common text editing tasks. This drives the need for fine-tuning on task-specific datasets. The impact of this task-specific data augmentation for text editing tasks has already been shown in Kim et al. (2022). For this work, we compare our task-specific fine-tuned models against their FLANT5 un-tuned counterparts referred to as FLANT5-XL (Table 3(b)). We see a substantial gap

___
[8] task was one of gec, simplify, clarify, coherence, formalize, neutralize and paraphrase

| CoEdIT-xl | GPT3-Edit | Tie | Neither |
|:---:|:---:|:---:|:---:|
| 64% | 10% | 4% | 22% |

Table 4: Human evaluation results: Pair-wise comparison of CoEdIT-xl against the best-performing 175B-parameter instruction-tuned LLM for text editing (GPT3-Edit). Scores indicate the % of test inputs for which the human annotators preferred the said model.

between the two for all datasets and model sizes, thus, confirming prior findings.

**Quality of Instructions.** While we developed with a limited set of task-specific instructional prompts, there has been widespread work on the prompt sensitivity of LLMs, especially with growing model capacity (Lu et al., 2022). To assess the robustness of CoEdIT models on instructional prompts, we train another baseline CoEdIT-xl model with randomized task-specific instructions (henceforth referred to as CoEdIT-xl-r). Specifically, the entire training dataset was randomized, where an instruction from one task was replaced randomly by an instruction from another task. Table 3(c) shows the results for this experiment. We observe that while CoEdIT-xl-r achieves scores that are higher than the non-task-specific tuned FlanT5-xl (especially on edit-based metrics such as SARI), it significantly falls behind CoEdIT-xl on those, as well as on the style accuracy metrics such as formality transfer accuracy and paraphrasing semantic similarity. This indicates that while the instructional structure of the inputs and task-specific training makes the model learn how to make edits (which drives up the SARI scores), however, the accuracy of those edits suffers since they are trained with the wrong instructions most of the time. Overall, the improvements highlight the positive impact of task-specific training, and the gaps in performance highlight the negative impact of lack of proper instruction tuning.

## 6 Qualitative Results

We now address **RQ2** and **RQ3** (Section 4). We show that CoEdIT shows generalization abilities to adjacent tasks not seen during fine-tuning and can generalize to composite instructions containing a combination of tasks. Further, our human evaluation studies show that expert human evaluators find the text generated by CoEdIT to be of higher quality than a much larger instruction-tuned LLM.

|  | Sentence Compression | Politeness Transfer |
|---|:---:|:---:|
| **Model** | $\textbf{SARI}^{\uparrow}$/ $\textbf{CR(\%)}^{\uparrow}$ | $\textbf{S-BLEU}^{\downarrow}$/ $\textbf{TA(\%)}^{\uparrow}$ |
| GPT3-Edit | 23.98 / 6.09 | 63.31 / 63.11 |
| T5-xl (prefix) | 31.47 / 7.66 | 81.43 / 58.82 |
| FlanT5-xl | 33.21 / 15.29 | 91.91 / 52.69 |
| **CoEdIT-xl** | **35.17 / 22.78** | **60.32 / 64.45** |

Table 5: Comparison of CoEdIT-xl against the best-performing non-instruction-tuned model (T5-xl), non-task-specific-tuned model (FlanT5-xl) and GPT3-Edit on out-of-domain generalization.

### 6.1 Text Editing Quality

Since text editing is often subjective, and automatic metrics are not always accurate in measuring if an instruction is satisfied, we conduct human evaluations for our model outputs by linguistic experts on 50 test inputs to ensure they meet the instructional constraints. Given the automatic evaluation results in Section 5, we compare our 3B-parameter CoEdIT-xl model against the largest comparable 175B instruction-tuned LLM for text editing GPT3-Edit. Specifically, we conducted a pairwise comparison: each annotator was shown an instructional input and outputs from both models (they were not aware which output was generated by which model). They were then asked to evaluate the fluency, accuracy, and meaning preservation of the edited texts and choose the higher-quality output ("neither" and "tie" are also valid options). We collect three annotations for each question and use the majority vote as the final judgment.

Table 4 shows the results of the evaluation. The annotators prefer our CoEdIT model for 64% of the inputs, whereas, for 10% of the inputs, GPT3-Edit's output is preferred. In 4% cases, both models produce equally good outputs, whereas, for 22% of the inputs, both models generate unacceptable outputs. Table 12 provides a side-by-side comparison of the outputs generated by the two models.

### 6.2 Generalizability to Adjacent Tasks

We analyze the generalization capabilities of our models by evaluating them on a few related tasks that do not exist in the fine-tuning data. Specifically, we chose two standard NLP tasks – sentence compression (SC) (Filippova and Altun, 2013) and politeness transfer (PT) (Madaan and Yang, 2021). It is noteworthy that while our models were not fine-tuned on these exact tasks, we chose them so that the models could still comprehend them based

| Model | Size | IteraTeR | Fluency | Clarity | Coherence | Style | | |
|---|---|---|---|---|---|---|---|---|
| | | ITERATER$^\uparrow$ | JFLEG$^\uparrow$ | ASSET$^\uparrow$ | DiscoFuse-Wiki$^\uparrow$ | GYAFC($^\uparrow$/$^\uparrow$) | WNC($^\uparrow$/$^\uparrow$) | MRPC($^\downarrow$/$^\uparrow$) |
| CoEdIT-XL | 3B | 36.6 | 64.5 / 60.7 | 42.2 | 80.5 | 55.1 / 98.3 | 70.4 / 48.8 | 21.3 / 99.6 |
| CoEdIT-XL-C | 3B | 36.5 | 65.1 / 61.3 | 42.0 | 74.8 | 55.9 / 97.2 | 69.7 / 48.5 | 20.7 / 98.8 |

Table 6: Results for composite prompt training on single-task performance. Scores follow exactly as Table 2.

| CoEdIT-XL-C | GPT3-Edit | Tie | Neither |
|---|---|---|---|
| 38% | 34% | 3% | 25% |
| **CoEdIT-XL-C** | **CoEdIT-XL** | **Tie** | **Neither** |
| 34% | 21% | 14% | 31% |

Table 7: Human evaluation results: Pair-wise comparison of CoEdIT-XL-C against GPT3-EDIT and equivalent CoEdIT-XL (with chaining pipeline). Human annotators preferred the said model for % of test inputs.

on other tasks they were fine-tuned on. We define them as being *adjacent* tasks, which still exist within the scope of existing tasks but have not been seen during fine-tuning (blue lines in Fig. 2).

Similar to the previous experiment, in addition to GPT3-EDIT, we compare CoEdIT-XL against the similarly-sized prefix-tuned (T5-XL) model and the non-task-specific trained FLANT5-XL model (same models as the ones used in Table 3 (a) and (b)). For evaluation, we curated a set of new instructional prompts geared towards both the new tasks (details in Appendix C). We evaluated the models on the respective test datasets from Filippova and Altun (2013) and Madaan and Yang (2021).

Table 5 shows the results of CoEdIT-XL against various models on the sentence compression and politeness transfer tasks. For SC, we report the SARI metric for rewrite quality and compression ratio (CR) for task-specific quality. For PT, we report Self-BLEU (Zhu et al., 2018) for the rewrite quality[9] and Transfer Accuracy (TA) for the task-specific quality. We observe that CoEdIT consistently outperforms other models on both tasks, which indicates its generalization abilities on these new and unseen adjacent tasks. It is noteworthy that GPT3-EDIT performs quite well out-of-the-box on PT, but not so much on the SC task.

## 6.3 Generalizability to Composite Instructions

Finally, we also explore the capability of our model to understand composite natural language instruc-

---

[9] We report Self-BLEU based on the original PT paper since there are no references provided in the dataset.

tions. Composite instructions are made up of a combination of tasks. For example, for the composite instruction, "*Make the text simpler, paraphrase it, and make it formal*", the model needs to simultaneously perform simplification, paraphrasing and formalization of the input sentence.

Since there is no publicly available dataset for composite instructions, we create the CoEdIT-COMPOSITE dataset by expanding the CoEdIT dataset to a total of 90k pairs. In addition to the single-task instructions, we use seven new combinations of instructions as part of our training set, with each composite instruction having either two or three tasks. Specifically, these are GEC-Paraphrasing, GEC-Simplification, GEC-Paraphrasing-Simplification, Formality-Paraphrasing, Formality-Simplification, Formality-Paraphrasing-Simplification, and Paraphrasing-Simplification (more details in §A). We then fine-tune the FLANT5-XL model on CoEdIT-COMPOSITE (referred as CoEdIT-XL-C). The training details are summarized in §D.

We evaluate CoEdIT-XL-C on both single and composite instructions. For the single instructions, we use the same evaluation setup as in Table 2 and find that the overall performance of CoEdIT-XL-C is on par with that of CoEdIT-XL (Table 6). This shows that training the model additionally on composite prompts has no negative impact on single-task performance.

For composite instructions, we conduct human evaluations since there is no standard test dataset available. We use three new task combinations in addition to the seven seen during training to evaluate the model's generalizability. These are Coherence-Paraphrase, Coherence-Simplify, and Coherence-Simplify-Paraphrase. Specifically, we conduct two sets of pairwise annotations (similar setup as the one in Section 6.1) comparing CoEdIT-XL-C with GPT3-EDIT and CoEdIT-XL (shown in Table 7) on 30 composite instructions. For a fair comparison against CoEdIT-XL, we pre-

pare a chaining pipeline[10] by decomposing composite instructions into a sequence of multiple single instructions and executing them one-by-one. In 38% of cases, experts show a preference for COEDIT-XL-C, compared to 34% for GPT3-EDIT. In 3% cases, both models are preferred equally, whereas, for 25% of the cases, none of them are preferred. The experts prefer COEDIT-XL-C for 34% of the cases versus 21% for the chaining baseline. Both outputs are preferred equally in 14% cases, whereas, for 31% of the cases, both models generate unacceptable predictions. Table 13 provides a side-by-side comparison of outputs generated by these models.

## 7   Conclusions

We present COEDIT – an open-sourced dataset and set of instruction-tuned large language models that can act as a writing assistant by following natural language instructions to perform various textual edits by removing, updating, or adding words, phrases, and sentences. COEDIT achieves state-of-the-art performance on multiple text editing benchmarks, spanning syntactic, semantic, and stylistic edit requirements. Through extensive experiments, we have shown that COEDIT is capable of further generalizing to unseen, adjacent, and composite instructions to perform edits along multiple dimensions in a single turn. In our human evaluations, we observe that COEDIT can assist writers with various aspects of the text revision process at scale by following natural language instructions.

## Limitations

Although COEDIT achieves state-of-the-art performance on multiple text editing benchmarks, we acknowledge some limitations to our approach and evaluation methods. Our task-specific fine-tuning (like most other works) mainly focuses on sentence-level editing tasks, and its effectiveness on much longer sequences of texts that are more appropriate to real-world editing settings remains to be seen. Additionally, our system mainly focuses on non-meaning-changing text edits, thus, which could potentially limit the utility of our model to more real-world scenarios where fact-based editing or corrections are needed. Another limitation of our

work involves prompt sensitivity. While we construct our inputs by randomly choosing from a pool of verbalizers for every task, we acknowledge that different prompts may induce better or worse edits, and as we evaluate each input with a random verbalizer, a fully controlled comparison for each available prompt across all models is not done. Furthermore, the prompting format was kept uniform across all evaluated models, whereas some models may perform better with a different prompting format. We plan to address this in future work. Finally, computing resource requirements could pose some difficulty in replicating the results (which we try to address by sharing our models publicly).

## Ethics Statement

Since our work mainly focuses on non-meaning-changing text edits, we are able to avoid many issues involving generating harmful text. Although, there is still a possibility of small meaning changes for stylistic tasks, we try to reduce the chance of hallucinations by constraining the generation to strictly edit tasks in order to reduce the chance of adding any new information, or perpetuating biases.

## Acknowledgements

We sincerely thank Alice Kaiser-Schatzlein, Robyn Perry, Maya Barzilai, and Claudia Leacock for providing their invaluable linguistic expertise and insightful feedback with the evaluations. We also thank Max Gubin, Leonardo Neves, and Vivek Kulkarni for their helpful suggestions.

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

## A    Training Dataset Description

In this section, we discuss the details of the datasets used to create our training datasets and also expand on the dataset creation pipeline. For both CoEdIT and CoEdIT-Composite, we use the following datasets:

**Fluency**: We use three prominent corpora for GEC: the NUS Corpus of Learner English (NUCLE) (Dahlmeier et al., 2013), the W&I-locness (Bryant et al., 2019), and the NAIST Lang-8 Corpus of Learner English (Tajiri et al., 2012), which is one of the largest and most widely used datasets for GEC.

**Clarity**: We split Clarity into two sub-tasks, one focused on Text Simplification, and the other category focused on the set of edits outside of Simplification. In total, we use five corpora for Clarity tasks: Four of them - the Newsela corpus (Xu et al., 2015), WikiLarge (Zhu et al., 2010; Woodsend and Lapata, 2011; Kauchak, 2013), WikiAuto (Jiang et al., 2020) and a subset from Parabankv2 corpus (Hu et al., 2019) focus on text simplification, and the last one comes from the Clarity split of IteraTeR (Du et al., 2022b).

**Coherence**: We use the DiscoFuse dataset (Geva et al., 2019), as it involves linking two given sentences as coherently as possible using edit operations such as inserting discourse connectives.

**Style**: Owing to the subjective nature of Style edits based on different sub-intentions (eg. conveying writers' writing preferences, including emotions, tone, and voice, etc.). We use the following datasets for making different stylistic edits to reflect those distinctions:

- **Formality**: We use Grammarly's Yahoo Answers Formality Corpus (GYAFC) (Rao and Tetreault, 2018) which is a parallel corpus of informal and formal sentence pairs from two different domains.

- **Neutralization**: We use WNC (Pryzant et al., 2020), a dataset from the Subjective Bias Neutralization task, where the objective is to remove or mitigate biased words to make sentences more neutral;

- **Paraphrasing**: For paraphrase generation, we used the PARABANKV2 corpus (Hu et al., 2019), since it is a large-scale corpus that contains multiple diverse sentential paraphrases.

Once the raw datasets were collected, we randomly sampled them to the quantities mentioned in Table 1 based on a few heuristics such as old word retention, complexity ratios, dependency tree depth ratio, and character length ratio. The sampled pairs were then modified by prefixing the source texts with task-specific verbalizers (Appendix C) to convert a `<source, target>` pair to a `<instruction: source, target>` pair. All our models were then fine-tuned on the verbalized dataset.

**Composite instructions:** Table 8 shows the composition of the COEDIT-COMPOSITE dataset, in addition to the details about datasets and prompts. We use seven such composite instructions during model training. For the first three composite prompts (GEC-Paraphrasing, GEC-Simplification, GEC-Paraphrasing-Simplification), we use GEC datasets to extract datapoints that show simplification and paraphrasing edits in addition to GEC. For the next three prompts (Formality-Paraphrasing, Formality-Simplification, Formality-Paraphrasing-Simplification), we use the formality dataset (GYAFC) to extract pairs which exhibit paraphrasing and simplification edits in addition to formality. Lastly, for the last prompt (Paraphrasing-Simplification), we use the ParabankV2 paraphrasing dataset to extract data points which show a simplification of the source text in addition to paraphrasing.

To select the appropriate source-target pairs for a composite instruction, we use similar heuristics as with single-task instructions, i.e. old word retention, complexity ratios, dependency tree depth ratio, and character length ratio. For example, a source-target pair from a GEC dataset can be used for the composite instruction involving GEC, paraphrasing and simplification if the target and source sentence has a high edit distance and low complexity ratio, character length and word retention scores. The exact details can be found in the code.

Finally, for building the prompts for the composite instructions, we randomly sample from the task-specific verbalizers and concatenate them. The ordering of the single tasks in a composite instruction is also chosen randomly to ensure better generalization.

## B  Testing Dataset Description

Specifically, we consider the following datasets:

**Grammatical Error Correction**   We use the JF-LEG (Napoles et al., 2017) corpus of English sentences that represents a range of language proficiency levels and comprehensive fluency edits. For evaluation, we use the GLEU (Napoles et al., 2015) score as the primary metric and also report results using the SARI (Xu et al., 2016) metric.

**Text Simplification**   We use the TurkCorpus (Xu et al., 2016) and ASSET (Alva-Manchego et al., 2020) datasets, which were both created from WikiLarge data (Zhang and Lapata, 2017), where each complex sentence consists of multiple crowd-sourced reference simplifications. We report results using the SARI metric.

**Coherence**   We use the Coherence split of IT-ERATER (Du et al., 2022b), and the DISCOFUSE dataset (Geva et al., 2019), as it involves linking two given sentences as coherently as possible using edit operations such as inserting discourse connectives. We report results using the SARI metric.

**Iterative Text Editing**   We use ITERATER (Du et al., 2022b), an iterative text revision dataset spanning five edit intentions (Section 3) across three different domains (ArXiv, News, Wikipedia). We evaluate our models using the SARI metric. We report the performance on individual intentions – *Fluency*, *Clarity*, and *Coherence*, and also aggregated scores on the full dataset, which includes *Style* edits.

The rest of the section describes the evaluation setups for Style-related edits:

**Formality Style Transfer**   We use Grammarly's Yahoo Answers Formality Corpus (GYAFC) (Rao and Tetreault, 2018), a parallel corpus of informal and formal sentence pairs from two different domains. Similar to prior works, we evaluate the quality of rewriting using SARI, and the accuracy of style transfer using a formality classification model[11].

---

[11] https://huggingface.co/s-nlp/xlmr_formality_classifier

| Edit Intention | Datasets | Size | Example Input | Example Output |
|---|---|---|---|---|
| GEC-PARAPHRASE | NUCLE-14 Lang-8 BEA-19 | $1k$ | *Fix grammar in this sentence, and rewrite this sentence:* How about taking account of psychology? | One such perspective is to take psychology into account. |
| GEC-SIMPLIFY | NUCLE-14 Lang-8 BEA-19 | $1k$ | *Make this easier to understand, and remove grammatical mistakes:* So it was not like enjoying the tasteful gardens. | So it was not as if I could enjoy the pretty gardens. |
| GEC-PARAPHRASE-SIMPLIFY | NUCLE-14 Lang-8 BEA-19 | $1k$ | *Rewrite this sentence, change to simpler wording, and fix the grammar mistakes:* Due to ageing, some of the people may suffer from physical and mental depreciation. | Due to the effects of aging, some people may suffer. |
| FORMALITY-PARAPHRASE | GYAFC | $5k$ | *Make this sound more formal, and paraphrase:* writers dont think about what they will write, they just write!!! | Some writers can write freely without putting too much thought to it. |
| FORMALITY-SIMPLIFY | GYAFC | $2k$ | *Rewrite more formally, and make this text less complex:* Not to my knowledge...I'm a little curious myself now though. | I do not think so. |
| FORMALITY-PARAPHRASE-SIMPLIFY | GYAFC | $4k$ | *Rewrite the sentence to be simpler, make this sound more formal, and paraphrase this sentence:* my answer is what...very clever riddle!! | Your riddle was very clever, and I am unsure how to respond. |
| PARAPHRASE-SIMPLIFY | ParabankV2 | $5k$ | *Use simpler wording, and write a paraphrased version of the sentence:* In your second communication, you requested reinforcements. | You asked for backup in your second report |

Table 8: Example data instances with composite instructions in the COEDIT-COMPOSITE dataset (90K `<instruction: source, target>` pairs). Instructional prompts in the inputs are *italicized*.

**Neutralization** We use WNC (Pryzant et al., 2020), a dataset from the Subjective Bias Neutralization task. Based on prior works, we use Exact-Match (EM) for evaluations, which is the percentage of examples for which the edited text exactly matches the reference(s).

**Paraphrasing** We use the widely-used Microsoft Research Paraphrase Corpus (MRPC) (Dolan and Brockett, 2005), the STS benchmark from SemEval-2017 (STS) (Cer et al., 2017), and the Quora Question Pairs[12] (QQP) datasets. We evaluate paraphrasing on two criteria and metrics: Self-BLEU (Zhu et al., 2018) to measure the diversity of the paraphrases relative to the given source and reference texts, and Semantic Similarity[13] to measure meaning preservation.

## C Task Verbalizers

We manually curated a variety of task-specific verbalizers to construct the instructional inputs. Table 9 shows the full list of the verbalizers used for training and evaluations. Table 10 shows the verbalizers used for the experiments conducted in Section 6.2.

## D Training Details

We used the Adam optimizer with a learning rate of $1e - 4$. Each model is trained for 5 epochs

---

[12]https://quoradata.quora.com/ First-Quora-Dataset-Release-Question-Pairs

[13]We use the `paraphrase-mpnet-base-v2` model from SentenceTransformers (Reimers and Gurevych, 2019)

| Edit Intention / Task | Verbalizers |
|---|---|
| GEC | Fix grammar, Fix grammar in this sentence, Fix grammar in the sentence, Fix grammar errors, Fix grammatical errors, Fix grammaticality, Fix all grammatical errors, Fix grammatical errors in this sentence, Fix grammar errors in this sentence, Fix grammatical mistakes in this sentence, Fix grammaticality in this sentence, Fix grammaticality of the sentence, Fix disfluencies in the sentence, Make the sentence grammatical, Make the sentence fluent, Fix errors in this text, Update to remove grammar errors, Remove all grammatical errors from this text, Improve the grammar of this text, Improve the grammaticality, Improve the grammaticality of this text, Improve the grammaticality of this sentence, Grammar improvements, Remove grammar mistakes, Remove grammatical mistakes, Fix the grammar mistakes, Fix grammatical mistakes |
| Clarity | Clarify the sentence, Clarify this sentence, Clarify this text, Write a clearer version for the sentence, Write a clarified version of the sentence, Write a readable version of the sentence, Write a better readable version of the sentence, Rewrite the sentence more clearly, Rewrite this sentence clearly, Rewrite this sentence for clarity, Rewrite this sentence for readability, Improve this sentence for readability, Make this sentence better readable, Make this sentence more readable, Make this sentence readable, Make the sentence clear, Make the sentence clearer, Clarify, Make the text more understandable, Make this easier to read, Clarification, Change to clearer wording, Clarify this paragraph, Use clearer wording |
| Simplification | Simplify the sentence, Simplify this sentence, Simplify this text, Write a simpler version for the sentence, Rewrite the sentence to be simpler, Rewrite this sentence in a simpler manner, Rewrite this sentence for simplicity, Rewrite this with simpler wording, Make the sentence simple, Make the sentence simpler, Make this text less complex, Make this simpler, Simplify, Simplification, Change to simpler wording, Simplify this paragraph, Simplify this text, Use simpler wording, Make this easier to understand |
| Coherence | Fix coherence, Fix coherence in this sentence, Fix coherence in the sentence, Fix coherence in this text, Fix coherence in the text, Fix coherence errors, Fix sentence flow, Fix sentence transition, Fix coherence errors in this sentence, Fix coherence mistakes in this sentence, Fix coherence in this sentence, Fix coherence of the sentence, Fix lack of coherence in the sentence, Make the text more coherent, Make the text coherent, Make the text more cohesive, logically linked and consistent as a whole, Make the text more cohesive, Improve the cohesiveness of the text, Make the text more logical, Make the text more consistent, Improve the consistency of the text, Make the text clearer, Improve the coherence of the text |
| Formality Style Transfer | Formalize, Improve formality, Formalize the sentence, Formalize this sentence, Formalize the text, Formalize this text, Make this formal, Make this more formal, Make this sound more formal, Make the sentence formal, Make the sentence more formal, Make the sentence sound more formal, Write more formally, Write less informally, Rewrite more formally, Write this more formally, Rewrite this more formally, Write in a formal manner, Write in a more formal manner, Rewrite in a more formal manner |
| Neutralization | Remove POV, Remove POVs, Remove POV in this text, Remove POVs in this text, Neutralize this text, Neutralize the text, Neutralize this sentence, Neutralize the sentence, Make this more neutral, Make this text more neutral, Make this sentence more neutral, Make this paragraph more neutral, Remove unsourced opinions, Remove unsourced opinions from this text, Remove non-neutral POVs, Remove non-neutral POV, Remove non-neutral points of view, Remove points of view, Make this text less biased |
| Paraphrasing | Paraphrase the sentence, Paraphrase this sentence, Paraphrase this text, Paraphrase, Write a paraphrase for the sentence, Write a paraphrased version of the sentence, Rewrite the sentence with different wording, Use different wording, Rewrite this sentence, Reword this sentence, Rephrase this sentence, Rewrite this text, Reword this text, Rephrase this text |

Table 9: Complete list of task-specific verbalizers used in our training and test datasets.

| Edit Intention / Task | Verbalizers |
|---|---|
| Sentence Compression | Shorten the sentence, Shorten this sentence, Compress this sentence, Shorten this text, Compress this text, Write a shorter version for the sentence, Rewrite the sentence to be shorter, Rewrite this sentence in a shorter manner, Rewrite this sentence for shorter length, Make the sentence short, Make the sentence shorter, Make this shorter, Shorten, Compress, Shorten this paragraph, Shorten this text |
| Politeness | Increase politeness, Make this polite, Make this more polite, Make this sound more polite, Make the sentence polite, Make the sentence more polite, Make the sentence sound more polite, Write more politely, Rewrite more politely, Write this more politely, Rewrite this more politely, Write in a polite manner, Write in a more polite manner, Rewrite in a more polite manner |

Table 10: List of task-specific verbalizers used for generalizability experiments.

| | Model | Size | IteraTeR | | | Clarity | Coherence | Style (Paraphrasing) | |
|---|---|---|---|---|---|---|---|---|---|
| | | | ITERATER-FLU$^\uparrow$ | ITERATER-CLA$^\uparrow$ | ITERATER-COH$^\uparrow$ | TURK$^\uparrow$ | DiscoFuse-Sport$^\uparrow$ | STS($\downarrow$/$\uparrow$) | QQP($\downarrow$/$\uparrow$) |
| (a) | COPY | - | 31.9 | 28.6 | 30.8 | 26.3 | 30.5 | 39.6 / 100 | 30.1 / 100 |
| | T5-LARGE | 770M | 15.1 | 23.8 | 21.7 | 34.2 | 28.4 | 18.0 / 50.5 | 18.2 / 62.5 |
| (b) | T0[*] | 3B | 27.1 | 28.0 | 21.2 | 34.8 | 33.6 | 27.7 / 88.7 | 10.2 / 63.4 |
| | T$k$-INSTRUCT[*] | 3B | 22.3 | 20.9 | 21.2 | 32.3 | 29.1 | 12.3 / 49.5 | 19.1 / 79.6 |
| | T0++[*] | 11B | 37.9 | 30.2 | 27.5 | 34.1 | 35.1 | 29.9 / 99.0 | 16.6 / 79.1 |
| (c) | LLAMA | 7B | 32.4 | 28.8 | 31.4 | 27.2 | 30.8 | 1.4 / 14.4 | 1.5 / 35.8 |
| | GPT3 | 175B | 20.9 | 21.8 | 21.2 | 34.6 | 27.0 | 0 / 8.2 | 0 / 13.1 |
| (d) | ALPACA | 7B | 33.0 | 28.9 | 31.0 | 27.4 | 30.8 | 0 / 25.8 | 0 / 64.4 |
| | CHATGPT | - | 36.4 | 23.1 | 31.4 | 37.4 | 39.1 | 7.9 / 96.9 | 8.17 / 96.5 |
| | INSTRUCTGPT | 175B | 43.7 | 28.1 | 33.9 | 38.9 | 45.8 | 11.6 / 88.7 | 9.3 / 95.5 |
| | GPT3-EDIT | 176B | 48.3 | 31.8 | 34.2 | 34.7 | 48.0 | 14.0 / 86.6 | 6.4 / 94.5 |
| (e) | ALPACA (FS) | 7B | 33.0 | 28.9 | 31.0 | 28.4 | 32.1 | 0 / 39.2 | 0 / 63.4 |
| | GPT3 (FS) | 175B | 33.9 | 30.8 | 33.9 | 38.7 | 46.4 | 0 / 2.1 | 0 / 5.9 |
| | CHATGPT (FS) | - | 36.2 | 27.0 | 35.6 | 37.3 | 49.4 | 0 / 88.7 | **0 / 96.8** |
| | INSTRUCTGPT (FS) | 175B | 44.7 | 31.7 | **37.5** | **39.5** | 56.9 | **0 / 91.8** | 0 / 96.1 |
| (f) | ITERATER | 570M | 36.7 | 30.4 | 34.7 | 27.2 | 40.9 | 24.1 / 81.4 | 20.6 / 94.3 |
| | DELITERATER | 570M | 32.0 | 28.9 | 30.6 | 27.8 | 31.7 | 24.7 / 87.6 | 20.4 / 96.6 |
| | PEER-3B[*] | 3B | 51.4 | 32.1 | 32.1 | 32.5 | - | - | - |
| | PEER-11B[*] | 11B | **52.1** | **32.5** | 32.7 | 34.1 | - | - | - |
| (g) | **CoEdIT-L** | 770M | 46.8 | 30.9 | 31.5 | 38.5 | 70.5 | 22.4 / 92.8 | 15.7 / 97.2 |
| | **CoEdIT-XL** | 3B | 50.4 | 31.3 | 31.5 | 38.5 | 74.6 | 20.8 / 94.8 | 15.4 / 97.8 |
| | **CoEdIT-XXL** | 11B | 51.6 | 31.8 | 31.5 | 38.2 | **76.2** | 21.8 / 93.8 | 15.4 / 98.2 |

Table 11: Comparison of CoEdIT against various baselines (**on sub-tasks and additional datasets to Table 2**), divided into seven groups: (**a**) a copy baseline and T5-LARGE baseline prefixed only with task-specific tags (i.e. `<gec>`, `<clarity>`, etc.) (**b**) T5-based models, (**c**) Decoder-only LLMs, (**d**) Instruction-tuned LLMs, (**e**) Few-shot evaluations of large pre-trained models (**f**) SOTA text editing models, and, (**g**) Variants of CoEdIT models (our work). The first scores for each task (excluding STS and QQP style tasks) are SARI scores. For STS and QQP. the first scores are Self-BLEU scores, and the second is semantic similarity. ($\uparrow$) indicates higher is better, and ($\downarrow$) indicates lower is better. The best-performing models for each dataset are highlighted in bolded boxes. (FS) denotes few-shot evaluation.

with early stopping. All models were fine-tuned on A100 GPUs using Deepspeed (Rasley et al., 2020). Maximum sequence lengths for both the source and the target were set to 256 tokens (via filtering). The best-performing checkpoints were chosen based the validation loss.

# E Model Performance

Table 11 compares the performance of CoEdIT with the other models on the remaining test datasets. We observe similar trends as the ones observed in Table 2, where CoEdIT outperforms most models we compare against.

# F Data Examples

| Instruction | GPT3-EDIT | CoEdIT-xl (ours) |
|---|---|---|
| *Fix all grammatical errors:* Every one would expect to leave his owns and secrets dreams like to be a painter, a writter ... | Every one would expect to leave his owns and secrets dreams like the painting and the write ... | Every one would expect to leave his own and secret dreams like to be a painter, a writer... |
| *Make this simpler:* Fives is a British sport believed to derive from the same origins as many racquet sports. | Fives is a British racquet sport. | Fives is a British sport. It is thought to have come from the same place as many racquet sports. |
| *Fix coherence in this text:* Phantom Budo is an incredibly powerful ability. Spike has yet to fully develop this awesome fighting technique. | Phantom Budo is an incredibly powerful fighting technique. Spike is yet to fully develop it . | Phantom Budo is an incredibly powerful ability, but Spike has yet to fully develop it. |
| *Use different wording:* Judge Gerald W. Heaney, in dissent, said the authorities should have allowed the prisoner to be medicated without the consequence of execution. | Judge Gerald W. Heaney, who disagreed with the majority decision, said the prisoner should be medicated after important testimony is received. | In his dissent, Gerald W. Heaney said the authorities should have allowed the prisoner to be treated without execution. |

Table 12: A sample of text revisions generated by GPT3-EDIT and CoEdIT-XL (ours).

| Instruction | GPT3-EDIT | CoEdIT-XL (chaining) | CoEdIT-XL-C |
|---|---|---|---|
| *Remove all grammatical errors from this text, and make this text less complex:* They follow the way the Sun goes from the east to the west. | The Sun rises in the east and sets in the west. | They move with the sun as it goes from east to west. | They follow the sun as it goes from east to west. |
| *Make the sentence grammatical, rewrite the sentence with different wording, and make this text less complex:* Fives is a British sport believed to derive from the same origins as many racquet sports. | Fives is a British sport that is believed to derive from the same origins as many racquet sports. | It is thought to have come from the same sources as other racquet sports. | Fives is a British sport that is believed to have originated from racquet sports. |
| *Make the text coherent, and rewrite this text:* Nebulous is initially unconcerned when he is unable to finish his cheese sandwich. He soon discovers he is dealing with an infinite sandwich and immeasurable garnish. | Nebulous is initially unconcerned when he is unable to finish his cheese sandwich. He soon discovers he is dealing with an infinite sandwich and immeasurable garnish. He is then unable to finish his cheese sandwich. | Nebulous wasn't too worried about the fact that he couldn't finish his cheese sandwich, but soon realized that this was an endless sandwich and an endless amount of condiments. | Nebulous is initially unconcerned when he is unable to finish his cheese sandwich, but soon realizes he is dealing with an endless sandwich and an endless supply of condiments. |
| *Make the text more coherent, simplification, and rewrite this sentence:* Phantom Budo is an incredibly powerful ability. Spike has yet to fully develop this awesome fighting technique. | Phantom Budo is an incredibly powerful ability. It is an awesome fighting technique that Spike has yet to fully develop. | This fighting technique of Spike is still in development. | Spike has yet to fully develop his Phantom Budo ability. |
| *Rewrite the sentence to be simpler, and paraphrase this text:* The jury asked for transcripts of Quattrone's testimony about his role in the IPO allocation process. | The jury asked for transcripts of Frank Quattrone's testimony about his role in the initial public offering allocation process. | The jury requested a copy of Quattrone's testimony. | Transcriptions of Quattrone's testimony were requested by the jury. |
| *Rewrite more formally, and make this easier to understand:* brian, he asked the coolest questions, they suspended him though, i dont know what's up with that! | Brian, he asked the coolest questions, they suspended him though, I don't know what's up with that! He was a cool guy, and I liked his questions. | I do not understand why Brian was suspended. He had some great questions. | I do not know why they suspended Brian. |

Table 13: A sample of text revisions generated by GPT3-EDIT, CoEdIT-XL (chaining) and CoEdIT-XL-C for composite instructions.