# OpenReview forum: "CoEdIT: Text Editing by Task-Specific Instruction Tuning"
_EMNLP/2023/Conference — EMNLP 2023 Findings_

### Official Review · Reviewer_UcMe · 2023-07-27

**Soundness:** 2

**Excitement:**

3: Ambivalent: It has merits (e.g., it reports state-of-the-art results, the idea is nice), but there are key weaknesses (e.g., it describes incremental work), and it can significantly benefit from another round of revision. However, I won't object to accepting it if my co-reviewers champion it.

**Paper Topic And Main Contributions:**

This paper describes a text editing system based on more human-readable task instructions such as “make the sentence simpler” instead of fixed actions. The authors constructed a dataset CoEdit from public text editing dataset ITERATER by converting the editing intentions into human instruction triplets <instruction: source, target>. The dataset is then used to fine-tune varieties of FlanT5 models. These models are compared against a list of major editing systems and LLM models with both quantitative and qualitative evaluations on several other editing benchmarks. A human evaluation is also included to examine the performance of performing composite editing involving more than one editing task.

**Questions For The Authors:**

LN299, "... the input sentences and its corresponding revised reference were pre-pended to the instructional prompt". Simply adding these texts may not sound optimal as instruction prompts, and redundant for well-understood task like text-editing. Have you tried to only give the same triplets as you did in CoEdit dataset to these models? e.g., <please simply the text, source, target>

**Reasons To Accept:**

Rather extensive benchmark comparison is included in this paper with all major editing systems. The introduced model achieves competitive results with them while maintaining much smaller model size. The author conducted experiments on composite text editing by human evaluation, which has been less discussed in previous works.


**Reasons To Reject:**

This major change in the proposed work is on fine-tuning text editing models using natural instruction instead of plain prefix, and thus using the newly constructed CoEdit dataset. I may have missed it but Section 3 didn’t include further details about the model training based on previous work of FlanT5. It is unclear then what contributes most to the performance variations, and especially in comparing with general purpose LLMs on one specialized task. The more comparable work of PEER (Schick et al., 2023) also tackles slightly wider range of problems as the author also pointed out in LN139. Since instruction based tuning has been explored in many works, the novelty at this point can be limited.

In the meantime, the effort implied in the dataset construction is less significant, as it is constructed from public ITERATER+  and as said in LN077 “based on rules that introduce lexical and semantic variations”.

**Reproducibility:**

3: Could reproduce the results with some difficulty. The settings of parameters are underspecified or subjectively determined; the training/evaluation data are not widely available.

**Reviewer Confidence:**

3: Pretty sure, but there's a chance I missed something. Although I have a good feel for this area in general, I did not carefully check the paper's details, e.g., the math, experimental design, or novelty.

**Typos Grammar Style And Presentation Improvements:**

The CoEdit has been reused in different places to refer to both the model and dataset. This can be confusing in the reading and it may be better to have more clear indications.

---

> ### Author Rebuttal · Authors · 2023-08-29
>
> We appreciate your time and effort in reviewing our submission. We acknowledge your feedback and comments. The following is our response:
>
> **Details of model training and source of performance variations**
>
> We have shared the details of training datasets, evaluation datasets, evaluation metrics, task verbalizers, and training configurations in Appendixes A-D. Appendix E has details on example data points (showing the input and outputs of different models).
> In terms of the sources of improvements, we show it via our ablation studies. Specifically, there are two main sources of improvement:
> - Instruction-tuning (vs. prefix-tuning) and the underlying FlanT5 models
> - Task-specific training (via the high-quality human-annotated datasets and the dense task space)
>
> **Novelty/Differences against PEER**
>
> While we have highlighted the overlap with PEER in the related work, we have also described how there is a significant difference in the two works (L131-144).
>
> Our work is focused on text editing and revision (different problem scopes), uses task-specific datasets and instruction-tuning models (different modeling approaches), and further investigates different parts of the problem space (adjacent tasks and compositional instructions).
>
> Moreover, we publicly release our code, data, and models – this is a crucial contribution to the community, as there do not currently exist any replicable, open-source models focused on general-purpose text editing that can perform text editing with high quality while being small-sized (3-13B parameters). PEER is neither as performant, nor publicly available.
>
>
> **Dataset Creation (L77)**
>
> We agree that the dataset creation in this work is not a significant amount of effort, as it is sourced from publicly available datasets, similar to many prior works. However, we are not claiming the dataset as a novel contribution, other than the fact that we release our datasets publicly. The focus of the paper was to improve text editing performance with task-specific instruction-tuned models.
>
> **Single-instruction prompting (L299)**
>
> The instructional prompts were constructed in a similar matter to the triplets within the CoEdit dataset and particularly taken inspiration from the few-shot setup in Brown et al. (2020):
> ```
> Poor English input: I eated the purple berries.
> Good English output: I ate the purple berries.
> Poor English input: Thank you for picking me as your designer. I’d appreciate it.
> Good English output: Thank you for choosing me as your designer. I appreciate it.
> ```
> that was utilized for correcting grammatical errors. We constructed our in-context examples in the format:
> ```
> <please simplify the text>: <source>
> Answer: <target>
> ```
> in which we gave four examples per intent that were prepended to the instructional prompt for each instance. We will include this description of the setup in the updated revision of the paper to clarify any confusion about the experimental setup under the few-shot setting.

---

### Official Review · Reviewer_MrqV · 2023-08-02

**Soundness:** 3

**Excitement:**

3: Ambivalent: It has merits (e.g., it reports state-of-the-art results, the idea is nice), but there are key weaknesses (e.g., it describes incremental work), and it can significantly benefit from another round of revision. However, I won't object to accepting it if my co-reviewers champion it.

**Paper Topic And Main Contributions:**

This work proposes to use fine-tuned large language model to approach text editing task.
It conducts extensive evaluations comparing with baselines including existing text-editing models, instruction-tuned LLMs and general-purpose LLMs, showing edited results generated by the proposed method have better quality.

**Questions For The Authors:**

1. What's the detail about the few-shot learning? E.g., how many samples are selected; necessary token length change after few-shot setting, etc.
2. I feel confusing about the term 'prompt-tuned' large pre-trained models. What do you mean by this?


**Reasons To Accept:**

1.The proposed method is able to achieve better results with less parameter on text-editing task.
2. The comparison between the proposed method and existing methods is extensive and showing strong support to the claimed contribution in the paper.
3. There is some insight discussion about the general-purpose and task-specific tuning regarding to text-editing task.
4. The implementation details including the prompts are clearly described in the paper, making it easier for others to reproduce.


**Reasons To Reject:**

1. The novelty is limited due to the scope of the idea. While fine-tuning LLM can be applied to different tasks, the proposed method aligns with the typical approach within this paradigm. Consequently, the novelty is restricted by the commonly referred method and the limitations of the particular task.



**Reproducibility:**

4: Could mostly reproduce the results, but there may be some variation because of sample variance or minor variations in their interpretation of the protocol or method.

**Reviewer Confidence:**

4: Quite sure. I tried to check the important points carefully. It's unlikely, though conceivable, that I missed something that should affect my ratings.

---

> ### Author Rebuttal · Authors · 2023-08-29
>
> We appreciate your time and effort in reviewing our submission. We acknowledge your feedback and comments. The main feedback is in terms of novelty, and the following is our response:
>
> In terms of novelty, we believe our work is novel/valuable to the community in two main ways:
>
> **Empirical novelty**
> - Our work is the first to gather task-specific datasets, leverage instruction-tuning, and show that a single model can achieve state-of-the-art results on multiple text editing benchmarks.
> - The related works (covered in Section 2) are either not based on instruction tuning, use different modeling techniques such as tag-based sequence labeling, or are not general enough to work on multiple text editing tasks.
>
>
> **Practical novelty**
> - Our models are over 60x smaller than the other publicly available ones we compare against, and achieve a better performance. This is impactful for numerous reasons – it will make writing assistants easier to deploy, and cut down serving costs, which are prohibitively expensive.
> - We release our code, data, and models publicly – this is a crucial contribution to the community, as there do not currently exist any replicable, open-source models focused on general-purpose text editing that can perform text editing with high quality while being small-sized (3-13B parameters). The closest one to our work (PEER) is neither as performant, nor publicly available.
>
> **Response to questions**
>
> We conducted the few-shot experiments under a 4-shot setting to account for the maximum input length of the smallest model tested. We randomly selected four examples per intent from the CoEdIT training dataset, which were prepended to the instructional prompt as part of the input in the format:
> ```
> <prefix>: <source>
> Answer: <target>
> ```
> where we took specific inspiration from the few-shot experiments in Brown et al. (2020) that prepended examples to the input in the format:
>
> ```
> Poor English input: I eated the purple berries.
> Good English output: I ate the purple berries.
> Poor English input: Thank you for picking me as your designer. I’d appreciate it.
> Good English output: <completion>
> ```
>
> As for the term ‘prompt-tuned,’ we utilized this term and ‘few-shot’ interchangeably for describing and showing examples of good revisions as part of the instructional input.
>
> We will replace all occurrences of “prompt-tuned” with “few-shot” for uniformity across the paper and clarity of terminology. We will also make the details of these experiments clearer.

---

### Official Review · Reviewer_NJjA · 2023-08-11

**Typos Grammar Style And Presentation Improvements:** 1. In section 4.1 the heading of the …
**Soundness:** 3

**Excitement:**

3: Ambivalent: It has merits (e.g., it reports state-of-the-art results, the idea is nice), but there are key weaknesses (e.g., it describes incremental work), and it can significantly benefit from another round of revision. However, I won't object to accepting it if my co-reviewers champion it.

**Paper Topic And Main Contributions:**

In this paper, the author fine-tuned a LM, Flan-T5 (L, XL, XXL) with a set of text editing dataset and instructions.
The author explored two main research aspect 1) How good can such a text-editing instruction fine-tuned LM perform. 2) Can such a model be generalised to unseen text-editing task or compositional text-editing task?

According to the author, the **contributions** of this work are:
1. It fine-tuned a LM with various text-editing tasks and achieved SOTA performance on most of them.
2. The purposed fine-tuned LM is parameter-efficient compared with other text-editing models.
3. The purposed model has good generalisation performance on both unseen and compositional text-editing tasks.
4. The fine-tuned model and the data used will be released.
(However, I have concerns towards point 2 and 3. Please see the Reasons to Reject below.)

The training data are a set of public text editing dataset, covering tasks of fluency, coherence, simplification, paraphrasing, formalising and neutralising. The author mainly focus on non-meaning-changing editing tasks. The author rewrites the instruction prompts to provide variation.

The author conducts **3 main experiments**:
- Text editing performance on the test set of the fine-tuning dataset. The purposed model is compared with a series of baseline models, including zero-shoting or few-shoting. Human evaluation on comparing the purposed model with GPT3-Edit model.
- Ablation studies to investigate the effects of 1) general instruction tuning, 2) text-editing tasks training 3) instruction quality.
- Generalisation performance on unseen and compositional text editing tasks.



**Questions For The Authors:**

1. From the paper, I'm not sure how exactly did the author paraphrase the instruction prompts to provide variations. Is it manually paraphrased or automatically? What's the policy when doing so? How many variations are there for each task?
2. The observation on Line 440 is interesting, but what exactly do you mean by randomised task-specific instructions? Do you keep the same random instructions for all data in each task? Or they are random within each task? If it is the latter one, I think you need clearer explanation on this observation.
3. In line 065, you claim "Our experiments demonstrate that fine-tuning instructions for specific tasks is more effective than multi-task fine-tuning", I believe by "multi-task fine-tuning models" you mean models like ChatGPT and ALPACA. It might be unfair to make such a claim if not having a clear control on the size of the instruction-tuning dataset. Furthermore, in Figure 2, from a reader's perspective, it seems you are suggesting the left general-purpose cannot generalise to composite and unseen tasks.

**Reasons To Accept:**

1. The paper has shown instruction-tuning can effectively push a range of text-editing tasks into one LM with good performance. Extensive experiments are done to support the conclusion.
2. The paper mentioned an interesting and important topic, generalisability of the instruction tuning. However, the experiments setting is a bit weak.

**Reasons To Reject:**


1. Key baseline is missing in the performance experiment:
    The author has compared with a series of baseline models in Table 2. However, it could be important to compare with single text-editing task fine-tuned Flan-T5. This could illustrate if the multi-text-editing-task instruction tuning can effectively improve the performance.

2. The argument on the performance on generalisation is a bit weak:
    - For the experiments on generalisability to unseen tasks, the task choices are sentence compression and politeness transfer. I have concern that the sentence compression is very similar to the text simplification task, which is used as training data. The politeness transfer task also has similarity with the formalising task. If you think there are clear differences, please explain the similarity and difference a bit more, or if possible choose a different task as unseen task.
    - The way you collect the training data for compositional task, mentioned in line 1130, is selecting special cases from the current GEC and GYAFC training data. If they are compositional enough are questionable.
    - In line 558, you didn't tell how many new task combinations did you use to evaluate the compositional generalisability. Furthermore, the test results in Table 7 are only from 30 composite instructions.

**Reproducibility:**

4: Could mostly reproduce the results, but there may be some variation because of sample variance or minor variations in their interpretation of the protocol or method.

**Reviewer Confidence:**

4: Quite sure. I tried to check the important points carefully. It's unlikely, though conceivable, that I missed something that should affect my ratings.

---

> ### Author Rebuttal · Authors · 2023-08-29
>
> We sincerely appreciate your time and effort in reviewing our submission. We acknowledge your insightful feedback and comments, and will address your suggestions, and make the following updates in our current submission:
>
> **Single-task Baselines**:
> In this work, our focus was on multi-task text editing models – where multiple text editing tasks can be performed with a single model. Kim et al. (2022) have demonstrated the benefits of training multi-task text editing models over single-task text editing models in their DELIteraTeR system. While they did not specifically use instruction tuning, other works have observed similar results (e.g., Sanh et al., 2022). Hence, we chose to skip it in the interest of space and the novelty of other results. However, we will address this by adding an additional baseline focusing on single-task text editing models in the final version.
>
> **Generalization**:
> The reviewer’s observation is accurate, and this was a deliberate choice. We have explained this in lines 499-505:
>
> > “It is noteworthy that while our models were not fine-tuned on these exact tasks, we chose them so that the models could still comprehend them based on other tasks they were fine-tuned on. We define them as being adjacent tasks, which still exist within the scope of existing tasks but have not been seen during fine-tuning (blue lines in Fig. 2).”
>
> By generalization, we mean that the model was able to generalize to “adjacent text editing tasks,” where we have defined adjacent tasks in L502. Studying inter-task dependency is out of the scope of this work, but we agree with the reviewer that the phrasing of “generalizability” here can be misconstrued. We will rename the subsection from “generalizability on out-of-domain tasks” to “generalizability to adjacent text editing tasks” for better clarity.
>
> **Compositional Task Data**
> We agree with the reviewer's observations about how we built the training dataset and that it does not reflect all possible cases of compositionality (since we chose only 7/120 possible combinations). This was a deliberate choice as there does not exist any compositional dataset for text editing, and collecting human-annotated datasets is expensive. Therefore, we leveraged the potential overlaps between tasks and used task-specific metrics to create compositional and multi-task datasets from single-task datasets. Thus, our aim with this experiment was to study compositionality within text editing and show that even with this simple setup, it is possible for much smaller models to outperform larger general-purpose LLMs. In the future, we will look into studying broader compositionality setups with higher-quality data.
>
> As to **how we created the compositionality data**, we used three new task combinations (in addition to the seven used during training) to evaluate the compositional generalizability in our experiments. These are Coherence-Paraphrase, Coherence-Simplify, and Coherence-Simplify-Paraphrase. Due to space constraints, we mention this information in the appendix (L1150). We will include this in the main body of the paper in the revised version.
>
> Regarding **using 30 composite instructions** in Table 7 for evaluation, we acknowledge this feedback. Our decision was based on the fact that there is a lack of an existing standard evaluation dataset to measure compositional generalizability and the challenges related to the time and costs associated with human evaluation.
>
> ***Responses to questions***:
>
> **Instruction Prompt Creation**: We have provided the full set of instruction prompts in Table 9. These were created heuristically by using an initial list of instructions that were paraphrased using rules to ensure lexical and semantic diversity. The final list was then chosen by manual filtering of the candidates. We will release these scripts in our code for full reproducibility.
>
> By **randomized Task-Specific Instructions**, we mean that we combine all instructions (from all tasks) into a list and then randomly sample an instruction to be prepended to a source text. In this way, it is possible that a simplification data point might get an instruction for grammatical error correction, and vice-versa, and so on. Hence, the instructions are randomized across the entire dataset.
>
> **Line 065**: We acknowledge the feedback on this point. With our current phrasing, we are not saying that “left general-purpose cannot generalize to composite and unseen tasks”; we are instead saying that task-specific (right) can generalize better to composite and unseen tasks since they are closer in the “task space.” We will ensure we edit the paper's text to convey the right meaning.
> We will also fix the other issues pointed out by the reviewer pertaining to naming the “Baseline” to “Copy Baseline” and adding PEER’s base model.

---

### Meta-Review · Area_Chair_soee · 2023-09-22

**Recommendation:** 3

**Metareview:**

The paper introduces LLMs that are able to perform multiple text editing tasks, while beating strong baselines, using fewer parameters and being faster than existing models. The training approach is based on fine-tuing Flan-T5 models of different sizes using a large set of instructions.

Reviewers praised the extensiveness of the experiments presented, comparing their models with several benchmarks, which allow to trust the conclusions related to performance. Other strengths of the paper include an analysis of the performance of the models in adjancent tasks and in composite tasks, including human assessments on output samples.

The main concern raised by reviewers is related to the novelty of the proposed approach compared to existing work (e.g. PEER), and the fact that fine-tuning LLMs is a common technique in the area. The authors' rebuttal has clarified where the novelty of their work lays, which includes practical reasons (e.g. inference speed, model size, etc.). Authors are also encouraged to include clarifications regarding "generalisability" since "adjacent tasks" would be more suitable than "out-of-domain", as they have also acknowledged.

---

### Decision · Program_Chairs · 2023-10-07

**Decision:**

Accept-Findings

**Comment:**

The paper introduces LLMs that are able to perform multiple text editing tasks, while beating strong baselines, using fewer parameters and being faster than existing models. The training approach is based on fine-tuing Flan-T5 models of different sizes using a large set of instructions.

Reviewers praised the extensiveness of the experiments presented, comparing their models with several benchmarks, which allow to trust the conclusions related to performance. Other strengths of the paper include an analysis of the performance of the models in adjancent tasks and in composite tasks, including human assessments on output samples.

The main concern raised by reviewers is related to the novelty of the proposed approach compared to existing work (e.g. PEER), and the fact that fine-tuning LLMs is a common technique in the area. The authors' rebuttal has clarified where the novelty of their work lays, which includes practical reasons (e.g. inference speed, model size, etc.). Authors are also encouraged to include clarifications regarding "generalisability" since "adjacent tasks" would be more suitable than "out-of-domain", as they have also acknowledged.